# Evaluation of Replacement Hearing Aids in Cochlear Implant Candidates Using the Hearing in Noise Test (HINT) and Pupillometry

**DOI:** 10.3390/audiolres15010013

**Published:** 2025-01-28

**Authors:** Yeliz Jakobsen, Kathleen Faulkner, Lindsey Van Yper, Jesper Hvass Schmidt

**Affiliations:** 1Research Unit for ORL—Head & Neck Surgery and Audiology, Odense University Hospital, 5000 Odense, Denmark; jesper.schmidt@rsyd.dk; 2Department of Clinical Research, Faculty of Health Sciences, University of Southern Denmark, 5000 Odense, Denmark; lvanyper@health.sdu.dk; 3Open Patient data Explorative Network (OPEN), Odense University Hospital, 5000 Odense, Denmark; 4Oticon A/S, 2765 Smørum, Denmark; ktsl@oticon.com

**Keywords:** hearing aid, cochlear implant, bimodal solution, signal-to-noise ratio, Hearing in Noise Test, pupillometry, speech performance, task engagement

## Abstract

Background/Objectives: Advances in cochlear implant (CI) technology have led to the expansion of the implantation criteria. As a result, more CI candidates may have greater residual hearing in one or two ears. Many of these candidates will perform better with a CI in one ear and a hearing aid (HA) in the other ear, the so-called bimodal solution. The bimodal solution often requires patients to switch to HAs that are compatible with the CI. However, this can be a challenging decision, not least because it remains unclear whether this impacts hearing performance. Our aim is to determine whether speech perception in noise remains unchanged or improves with new replacement HAs compared to original HAs in CI candidates with residual hearing. Methods: Fifty bilateral HA users (mean age 63.4; range 23–82) referred for CI were recruited. All participants received new replacement HAs. The new HAs were optimally fitted and verified using Real Ear Measurement (REM). Participants were tested with the Hearing in Noise Test (HINT), which aimed at determining the signal-to-noise ratio (SNR) required for a 70% correct word recognition score at a speech sound pressure level (SPL) of 65 dB. HINT testing was performed with both their original and new replacement HAs. During HINT, pupillometry was used to control for task engagement. Results: Replacing the original HAs with new replacement HAs after one month was not statistically significant with a mean change of SRT70 by −1.90 (95% CI: −4.69;0.89, *p* = 0.182) dB SNR. Conclusions: New replacement HAs do not impact speech perception scores in CI candidates prior to the decision of cochlear implantation.

## 1. Introduction

The eligibility criteria for cochlear implant (CI) surgery have been expanded due to advanced CI technology, allowing candidates with a greater degree of residual hearing (especially in the ear not considered for implantation) to benefit from these devices. Many of these candidates achieve better auditory outcomes when combining a CI in one ear with a hearing aid (HA) in the other ear—a strategy known as a bimodal solution [1].

This approach often necessitates switching to a HA that is designed to work in better alignment with the CI [2]. For many bimodal patients, the HA and the CI are fitted separately, often by a different clinical provider, and are not optimized to work together. For patients with severe hearing loss in the non-implanted ear, the consequence of having a standard HA in their non-implanted ear is likely minimal. However, for those with more usable residual hearing in the HA ear, such as those with asymmetric hearing losses, there is a greater need for an alignment of the acoustic and electric processing using a dedicated HA-fitting strategy [2].

Without proper amplification in both ears, patients may not achieve their maximum auditory performance [3]. Bimodal hearing, furthermore, leads to better auditory outcomes when the HA is properly fitted and synchronized with the CI, for example, alignment in frequency bandwidths and loudness growth. Switching to HAs compatible with the CI can provide improved speech recognition in noise and sound localization and supports better post-operative rehabilitation and quality of life [4].

In addition, modern HAs compatible with CIs also enable seamless audio streaming to both devices, ensuring balanced sound for phone calls, music and media. Patients can benefit from direct bimodal streaming and volume control of the HA and CI using the same app on their phone. This unified control system simplifies device management, reducing cognitive load, improving the user experience. Centralized control also enhances patient satisfaction and promotes more effective device use, particularly in challenging listening environments [3,5]. However, replacing the HA can be difficult, especially when the patient is accustomed to the sound quality provided by older generation technology or different HA brands, which limit access to bimodal streaming and shared controls of the CI and HA simultaneously. It is unclear how switching to a CI-compatible HA impacts the overall hearing performance in CI candidates.

When evaluating a patient for CI candidacy, measuring the HA benefit is important. For example, in the UK, The National Institute for Health and Care Excellence (NICE) has listed guidelines for cochlear implantation and recommends that a unilateral CI is offered to patients with severe to profound deafness who do not receive adequate benefit from acoustic HAs [6]. For patients that have some usable residual hearing in the opposite ear, the patient likely will experience significantly more benefit from a bimodal solution [7].

When evaluating a patient’s benefit from HAs, it is necessary that the HAs are fitted and adjusted adequately. Outdated or poorly fitted HAs can provide an insufficient amplification, potentially leading to poor outcomes and premature CI recommendations [8]. For this reason, it is recommended to adjust or even switch HAs during evaluation to ensure that HAs provide the best possible support for speech intelligibility when evaluating CI candidacy [7,9].

Also, when considering a bimodal solution, thorough assessment of pre-implant HA performance is essential, not least because these CI candidates tend to have more residual hearing. Severe hearing-impaired (HI) listeners may experience significant benefits from using bilateral HAs. These benefits include enhanced sound source localization, word recognition and speech intelligibility [10]. CI should, therefore, only be considered when well-fitted, up-to-date HAs do not provide sufficient benefit [11]. In these candidates, cochlear implantation can restore speech perception in quiet listening environments. However, unilateral CI-users often experience difficulties localizing sound and understanding speech in noise. Studies have shown that wearing a contralateral HA can provide substantial advantages in terms of speech recognition in noise, spatial hearing, sound quality and functional everyday communication [1,8,12,13,14]. To maximize the benefits from the bimodal solution, recent fitting guidelines advise matching the HA and CI in pitch and loudness [1,3,4,12,13,15].

Despite these benefits, large variability exists in the subjective outcomes with such a bimodal fitting. One of the challenges many bimodal listeners face is related to listening effort, defined as the allocation of mental resources to overcome challenges while carrying out a listening task [16]. Bimodal solution may offer improvements in sound quality and reduced listening effort for very challenging noise conditions [17].

### 1.1. Listening Effort

HI listeners, and CI users in particular, often report increased sustained listening effort when attempting to understand speech with their hearing devices [18,19,20,21]. The amount of effort needed to understand speech increases further as the task becomes more difficult, such as with increasing levels of background noise [22,23]. The cognitive processes required to understand speech in such environments can cause fatigue and stress, ultimately leading to social withdrawal in some of these listeners [24].

Despite listening effort being a common complaint, measuring it has proven difficult. Many objective measures have been developed in an effort to quantify listening effort, including pupillometry.

### 1.2. Pupillometry

Pupillometry is the recording of changes in pupil size, which has been shown to be larger when listening to speech in challenging environments. As such, pupillometry is a promising autonomic indication of effort [25,26]. Pupillometric measures are the peak pupil dilation (PPD) and the peak pupil latency (PPL), among others [25]. PPD and PPL increase as listening effort and task difficulty rise, especially in patients with hearing loss. Hearing-impaired listeners face a greater cognitive load, resulting in more pronounced pupil dilation in demanding auditory situations, particularly when tasks require increased focus and effort to understand speech or sounds [18,27,28,29].

Pupillometry has also been used to quantify listening effort during speech-in-noise tests in HI listeners, showing increased pupil diameter indicating an enhanced allocation of resources to reach similar speech intelligibility scores compared to normal hearing (NH) controls [30]. In summary, as the task becomes more difficult, the pupil diameter increases with increasing effort, if the listener is engaged in the task. However, when the task becomes too difficult to be successful, listeners will disengage from the task and pupil dilation decreases.

As mentioned above, one of the most challenging tasks for HI listeners is understanding speech in background noise, and, therefore, the pupil reaction can be measured during an adaptive speech-in-noise task that aims to determine the signal-to-noise ratio (SNR) at which the listener achieves a certain percentage correct score [31]. Combining such speech-in-noise tests with pupillometry offers a view into the cognitive processing resources that the HI listener must use to maintain a certain performance level of speech understanding at a fixed SNR [32].

### 1.3. Hearing in Noise Test

Many speech-in-noise tests are presented at a fixed level and SNR, which provides a percentage-correct score. However, this means that each patient may put in a different level of effort to achieve their score, making comparisons between patients less reliable. By using an adaptive procedure, the noise level can be adjusted to find the point where each person achieves a fixed percentage-correct score. This method ensures that everyone is putting in a similar level of effort, providing a more fair and consistent way to assess speech recognition. Often, in research, adaptive speech testing will track the 50%-correct word-recognition score. However, this level of difficulty may be unsustainable, and people often disengage from such challenging tasks. This disengagement can also influence pupillometric responses, as overly difficult tasks may lead to pupil responses indicating a lack of engagement. Tracking a more moderate performance level, such as the 70% used in this study, better reflects real-world listening conditions and is more suitable for assessing engagement through pupillometry.

The SNR for a certain percent-correct word-recognition score can be measured with the Danish Hearing in Noise Test (HINT) [33]. This test has been developed and validated for use in assessing speech recognition in noise of both NH and HI listeners [33]. HINT has been used to evaluate the listening performance in noise of CI users [34,35,36], as well as to assess the effectiveness of an HA [37].

This study uses HINT and pupillometry to evaluate the necessity of replacing HAs before deciding about unilateral cochlear implantation. As outlined above, bimodal listeners can benefit from features, such as direct streaming from a smartphone to both devices. To have access to these features, the HA and CI need to be compatible with one another, meaning that many bimodal candidates will need to use a different HA post-implantation. Thus, when considering unilateral CI in these patients, we need to ensure that speech intelligibility with the new replacement HA is not inferior to that with their original HAs. In this study, speech perception in noise was measured using HINT to compare the performance of CI candidates using either their original HAs or the replacement HAs that were compatible with the CI in a bimodal solution. As such, we evaluated if replacement HAs offer identical or potentially better speech intelligibility. Furthermore, speech perception in noise was measured under constant task engagement, as quantified by pupillometry.

### 1.4. Objective

This study investigated if changing HAs impacts speech perception scores in CI candidates prior to the decision of cochlear implantation on the poorer hearing ear. We aimed to determine whether speech perception in noise remains the same or improves with new replacement HAs compared to the patients’ original HAs.

The primary objective was to compare speech perception in noise using either original HAs or new replacement HAs that were compatible with a CI in the bimodal solution. To achieve this goal, HINT was used, and constant task engagement was ensured and controlled for using pupillometry.

## 2. Materials and Methods

### 2.1. Participants

Fifty bilateral HAs users (mean age 63.4; range 23–82) native Danish speakers with post-lingual hearing loss were enrolled in this study. All were candidates for a CI and users of HAs for at least a year. None of the participants had a history of ear or eye surgery. All participants provided written informed consent before starting the experiment.

This study was the initial part of a prospective randomized controlled trial [38] based on a single center, conducted at Odense University Hospital, Denmark, which was successfully registered at ClinicalTrials.gov (registration number: NCT04919928) and which has been approved by the Research Ethics Committee of Southern Denmark (Project-ID: S-20200074G) from 21 August 2020 to 31 December 2025. This part of the trial investigated whether changes in speech performance occurred after the replacement of HAs as a part of the pre-operative evaluation determining CI candidacy. The enrolment of participants started 1st February 2022.

### 2.2. Apparatus and Procedure

After enrollment, participants underwent testing using the Hearing in Noise Test (HINT) and pupillometry, wearing their original HAs. HINT lists consist of 20 sentences, and the task for the participant is to listen to the sentence and repeat what was heard. The HINT setup consisted of three loudspeakers (Fostex 6301NX Powered Monitor (Single) Mega Audio GmbH 55444 Waldlaubersheim, Germany) each positioned one m from the participant. Speech was delivered through a front-facing speaker, while the background noise masker was presented from two speakers positioned at a 45-degree angle to either side. The sound pressure levels from each speaker were regularly calibrated to 65 dB using a B&K 2636 sound level meter and the calibration function in the software (v. 4.0, Oticon, Smørum, Denmark).

All tests were conducted with participants using HAs in the best-aided condition.

The HINT testing took place in a standard clinical room with a background noise level of 37–42 dB, to provide a more realistic listening environment similar to everyday situations.

Every visit consisted of 2 test sessions. The order of the sentences in each test list was randomized before being presented. Prior to testing, participants received verbal instructions on how to listen and repeat the heard sentence and were encouraged to guess if they were if they were unsure of what they heard. Participants were asked to repeat the HINT sentences or as many words as they could understand or guess after the masking noise. The first session was conducted while the HINT was played at a fixed speech level of 65 dB, with the noise level adjusted adaptively to determine the speech reception threshold (SRT70)—the signal-to-noise ratio (SNR) at which participants achieved 70% correct word recognition. This SNR threshold was chosen over the typically 50% to make the task more realistic and easier to engage in for the patients, as more sentences would be heard and understood to keep motivation. The background noise consisted of multi-talker babble noise.

A new sentence list was used for each HINT session to prevent participants from recalling sentences between sessions. Participants responses were scored in real time by the examiner. The scoring was based on two metrics: the percentage of correct words out of a total of 100 words (5 words per sentence across 20 sentences) and the percentage of fully correct sentences out of 20 sentences.

The subsequent HINT was played using the determined SNR at SRT70 determined in the first HINT session. The speech signal was set at 65 dB, and the noise masker level was adjusted to each participant’s individual SNR, as determined in the first HINT session. If the adaptive procedure in HINT found an SRT70 below 20 dB SNR, this measured SRT70 value was used as the fixed level for the subsequent HINT session. However, if the adaptive procedure found an SRT above 20 dB SNR, the SRT70 was limited to +20dB SNR in the following session to ensure the noise onset and offset remained audible. This was confirmed verbally by the participant. The noise masker had to be greater than 45dB to ensure that the patient could detect the noise onset and offset to repeat the sentence at the right timeframe as illustrated in Figure 1.

During the HINT, participants wore a Pupil Labs Eye Tracker (v. 4.0, Oticon, Smørum, Denmark). The luminance of the room was kept at a constant of 20–40 lux using the “Philips Hue” app (Signify N.V. Eindhoven, Netherlands) and a light meter (RS Pro 180-7133, London, UK).

When performing pupillometry, the timing of noise onset/offset was very important. The measured SRT70 during the first session corresponded to the adaptive HINT. This test determined the participant’s ability to hear sentences or parts of sentences in the presence of a noise masker, adjusted to match the individual’s hearing ability. However, for conducting pupillometry in these severely hearing-impaired participants, the SRT70 had to be <20 dB. This ensured that the noise was detectable, allowing accurate measurement of pupil responses. When the participant heard the sentence, the pupil dilated and then returned to baseline as the noise ended. At this point, the participant was required to repeat the sentence, knowing exactly when the noise ended. During this task, the pupil dilated again due to the cognitive load of repetition. If the noise was undetectable, the participant might repeat the sentence immediately after hearing it, leaving insufficient time for the pupil to return to baseline. This would result in inaccurate measurements of pupil dilation during the repetition task (Figure 1).

Participants were asked to look forward and “relax” their gaze on the speaker in front of them. If the patient focuses their gaze on one place, it requires further cognitive effort [25]. For each measurement trial, the masking noise onset started three s prior to sentence onset and stopped three s after sentence offset, as demonstrated in Figure 1. Two seconds of silence were established before noise onset to allow for the pupil to return to the baseline level.

### 2.3. Replacement of Hearing Aids

The participants were then offered new replacement HAs (Phonak Link M or GN resound LiNXQuattro, ENZOQ) based on their personal preference and previous experience with these brands. These above-mentioned HAs are currently the only brands compatible with a Cis from Cochlear and Advanced Bionics in a bimodal solution. Bimodal solution enables synchronized hearing and direct streaming from various electronic devices, such as a smartphone or TV, simultaneously to the CI and HA.

The new replacement HAs were fitted according to the National Acoustics Laboratory formula-non-linear 2 (NAL-NL2) prescriptive fitting formula, to match international standard fitting rationale [39]. The HAs fittings were verified with Real Ear Measurement (REM) and adjusted accordingly as well as according to patient feedback to ensure optimally fitted HAs. The participants were offered additional adjustments if necessary.

Participants used the new replacement HAs for one month for the purpose of acclimatization and were randomized per protocol to either continuous use of HAs for an additional three months or referral to cochlear implantation after one month of using the new replacement HAs [38]. The participants were tested again with HINT and pupillometry after one month of acclimatization with the new replacement HAs. One participant passed away during the course of the study, prior to testing with the new replacement HAs, and five participants were satisfied with the new replacement HAs and therefore declined CI surgery. Five participants declined further participation in the project due to mental overload because they thought that the testing and questionnaires were too stressful. Half of the participants received the CI surgery after one month of acclimatization and the other half used the new replacement HAs for an additional three months before CI surgery.

### 2.4. Data Analyses

The pupillometry data was analyzed with PupilLabs acquisition and R2016a Matlab runtime environment to run. The Median Absolute Deviation Method (MAD) was applied because it is a robust method used for eye-blink detection [40].

Automatic blink removal removed additional data spanning from 35 ms before the blink to 100 ms after, by default. If more than 40% of the data were missing for one sentence due to blinks, data for that sentence were discarded and were not considered for further analysis.

The analyzed time range was set from four s to ten s, which is from noise onset to noise offset. The baseline was set from four s to 5.2 s, indicating the pupil stabilization and considering the pupil baseline.

The range for finding the PPD was set from 6.2 s to 9 s, indicating the sentence onset to sentence offset.

### 2.5. Statistics

The study hypothesized that speech perception would remain unchanged with new replacement HAs compared to the original HAs in CI candidates with residual hearing. A constrained linear mixed model was used to test this hypothesis, including the groups (original HAs, new HAs after one month, and new HAs after six months), time points (baseline and follow-up) and their interaction as fixed effects. Patient ID was included as a random effect to account for repeated measurements. Covariates such as age, sex and pure-tone average (PTA) thresholds for both the poorer-hearing ear (PTA-max) and the better-hearing ear (PTA-min) were also included as fixed effects.

## 3. Results

Seventy-seven participants were initially allocated at baseline. Fourteen were excluded because they either did not want to participate in the study or did not meet inclusion criteria. Sixty-three participants were enrolled, but before the one-month follow-up with new replacement HAs, three participants withdrew because they were satisfied, three participants withdraw without explanation, one participant had eye surgery and one participant switched HA brand. Five participants were excluded because they had missing data at the one month follow-up with new replacement HAs. Thus, 50 participants were tested one month after replacing the original HAs. As per protocol, a subset of participants (n = 25) continued using the replacement HAs for three months. Participant characteristics, SRT70 and pupillometry outcomes for the baseline and each of the follow-ups are described in Table 1.

The SRT70 ranged between 4 dB and 58 dB SNR.

When comparing the original HAs and new replacement HAs after one month of use across all participants, there was no significant improvement in the mean SRT70 of −1.90 dB SNR (95% CI −4.69;0.89, *p* = 0.182) (Figure 2a,b, Table 2).

Of note, 15 out of 50 participants showed an improvement of more than five dB SNR as the mean SRT70 when comparing original HAs with new replacement HAs after one month of use. SRT70 was significantly associated with PTA-min (that is, thresholds of the better-hearing ear) which increased the mean SNR of 0.36 (95% CI 0.24;0.49, *p* < 0.001) dB for every one dB increase in the hearing threshold on the better ear (Table 2).

Comparing the original HAs with new replacement HAs after three months within the control group, there was a change in mean SRT70 SNR of −0.67 (95% CI −4.31;2.98, *p* = 0.720) dB. This change was, however, not statistically significant (Figure 2a,b, Table 2).

In the second analysis of SRT70, all SRT70 values greater than 20 dB SNR were capped at a fixed value of 20 dB SNR. When comparing original HAs with new HAs after one month across all participants, there was a SNR change of −0.38 (95% CI −1.24;0.48, *p* = 0.387) dB though it was not statistically significant. When comparing original HAs with new HAs after one and three months within the control group, we found a change in the SNR of −0.74 (95% CI −1.99;0.51, *p* = 0.244) dB but it was not statistically significant (Table 2).

Analysis of the pupillometry outcomes revealed no statistically significant differences in PPD of 0.17 (95% CI −0.30;0.64, *p* = 0.476) mm or in PPL of −0.11 (95% CI −0.46;0.23, *p* = 0.510) seconds when comparing the original HAs with the new HAs after one month across all participants. When comparing the original HAs with the new Has after three months, within the control group, the change in mean PPD of 0.35 (95% CI −0.51;0.78, *p* = 0.346) mm and the mean change in PPL of −0.13 (95% CI −0.64;0.37, *p* = 0.610) seconds showed no statistical significance either. We conducted two separate analyses for PPD and PPL to evaluate whether pupillometric outcomes changed from the original HAs to the new HAs after one month, specifically for the two groups with SRT70 > 20 dB SNR and SRT70 ≤ 20 dB SNR.

In the SRT70 ≤ 20 dB SNR group, the change in PPD was 0.03 (95% CI: −0.02 to 0.09, *p* = 0.267) mm, which was not statistically significant. Similarly, in the SRT70 > 20 dB SNR group, the change in PPD was also 0.03 (95% CI: −0.005 to 0.06, *p* = 0.100) mm, with no statistical significance.

For PPL, the SRT70 ≤ 20 dB SNR group showed a change of 0.01 (95% CI: −0.02 to 0.05, *p* = 0.405) seconds, which was not statistically significant. In the SRT70 > 20 dB SNR group, the change in PPL was −0.0009 (95% CI: −0.02 to 0.02, *p* = 0.921) seconds, which was also not statistically significant.

Overall, the mean changes in both PPD and PPL were not statistically significant (Table 3; Figure 3a,b and Figure 4a,b).

There was, though, a significant association between PPD and PTA-max when comparing original HAs with new HAs after one month across all participants with 0.03 (95% CI: 0.01 to 0.05, *p* = 0.002) mm, as well as within the control group with 0.02 (95% CI: 0.004 to 0.04, *p* = 0.016) mm (Table 3).

## 4. Discussion

The aim of this study was to evaluate wether new replacement HAs provided equivalent or superior speech perception in background noise, as measured with the HINT compared to original HAs prior to CI surgery.

No statistically significant improvement was found in the mean SRT70 when comparing the original HAs versus the new replacement HAs with a mean change of −1.90 (95% CI −4.69;0.89, *p* = 0.182) dB SNR. However, fifteen out of fifty participants showed an improvement in SRT70 of at least five dB SNR after one month of using the new replacement HAs compared to their original devices. Some patients referred for a CI were not optimally fitted with the original HAs, meaning that they would have potentially achieved better speech perception had their original HAs been appropriately fitted. Among these patients, some utilized bilateral Contralateral Routing of Signals (biCROS) HAs, which, after using new replacement HAs, provided amplification to the previously unaided ear. Following the fitting of new HAs, these participants reported subjective improvements in sound localization and enhanced speech understanding. This observation suggests that new replacement HAs may improve speech intelligibility for certain patients, even though the changes were not statistically significant at the group level. These participants received amplification in the ear that had previously not been stimulated with sound. After the new HA fitting, these patients reported subjective improvements in sound localization and experienced enhanced speech understanding. This suggests that new replacement HAs may offer improved speech intelligibility at least for some patients, even though the improvement was not significant at the group level. Additionally, it is beneficial for patients to become accustomed to the new HAs before cochlear implantation, as the HA in the better-hearing ear will be paired with the CI in a bimodal solution. This allows patients to take advantage of features such as simultaneous streaming to both the CI and HA, enhancing their overall hearing experience.

The PPD and PPL did not show significant changes, indicating that participants maintained consistent task engagement when tested with both their original HAs and the new replacement HAs. This trend remained consistent even when an additional analysis was conducted with the study participants measured with SRT70 ≤ 20 dB to ensure that there was no statistically significant difference when leaving out subjects with SRT70 values above 20 dB SNR as well as study participants with SRT70 > 20 dB to ensure that there was no statistically significant difference when leaving out subjects with SRT70 values below 20 dB SNR.

Listening effort may be increased if the test difficulty increases. However, in the present study, the difficulty of the test was held constant by testing SRT70. If the participants did not hold a constant task engagement but experienced increased listening effort, we would have expected increased pupil dilation [28,30,41]. The fact that we did not find significant differences in pupil size suggests that task engagement was held constant throughout the HINT.

### Strengths and Limitations

A strength in our study is that task engagement was controlled for during the HINT and quantified with pupillometry.

When task demands increase, i.e., by decreasing the SRT70 in a speech in noise task, more cognitive resources are allocated causing high levels of effort. The cognitive resources are not unlimited and when the task becomes too hard, the participants quit and give up understanding the signal. The “quit” pattern is seen by the pupil’s immediate constriction. A similar pattern is seen when a task is too easy, requiring very little demand of cognitive resources, causing the listener to use no effort to complete the task [16]. Our study controlled for these factors and ensured adequate task engagement in our patient group.

Often patients are referred for CI surgery because their HAs are not meeting their hearing needs and impacting their social life [42]. This research was done to study the benefits of replacing the original HAs with new HAs before CI surgery. Although no significant improvement was observed when comparing the original HAs with new replacement HAs after one month, it is important to note that clinically relevant changes did occur in some individual subjects.

Many CI candidacy guidelines require severe-to-profound hearing loss to qualify for a CI, potentially excluding patients with moderate hearing loss who struggle with speech recognition in noisy environments. This can be regarded as a limitation, as current criteria may exclude patients from the study if they do not have severe-to-profound hearing loss, even though they could benefit from a CI [43]. Another important factor is the hearing ability of the better hearing ear. In our study, participants showed a wide range of hearing impairment in their better hearing ear, which contributed to the significant variation in SRT70 outcomes across the participants. All participants were CI candidates with severe hearing impairment in at least one year; however, some participants likely experienced a. lack of audibility provided by their HAs which resulted in very high SRT70 values where pupillometry could not be performed accurately, as it was impossible for the participants to detect the onset and offset of noise to correctly time their sentence repetition.

Similarly, the HINT consisting of 20 sentences induced great fatigue in some patients with poor speech comprehension causing poor word and sentence scores. This has been described in previous studies as a sign of giving up [30]. When the participants heard the HINT sentence, their pupils dilated and then returned to baseline as the noise ended. At this point, they were required to repeat the sentence, precisely timing their response to when the noise ended. During this repetition task, the pupil dilated again due to the cognitive load involved. If the noise were undetectable, participants might have repeated the sentence immediately after hearing it, leaving insufficient time for the pupil to return to the baseline, resulting in inaccurate measurements of pupil dilation during the repetition task.

The majority of our participants were severely hearing impaired, leading to very high SRT70 values. Excluding SRT70 values over 20 dB SNR would have reduced the sample size to a level insufficient for conducting mixed regression analysis. Therefore, it was necessary to implement a cut-off for SRT70 values exceeding 20 dB SNR at 20 dB SNR.

A further limitation in our study may be the high mean age of 63.4 years. The amplitude of pupil dilation in response to light decreases linearly with age because the age-related miosis (constriction of the pupil) has been attributed to degeneration of the dilator muscle in the iris [44]. Therefore, it can be more difficult to get a sufficient pupil response in older adults. Cognitive efforts still evoke pupil dilation in older adults; however, the overall amount of task-evoked pupil dilation is smaller in older adults [44].

## 5. Conclusions

The aim of this study was to test if it is of benefit for the patients to replace HAs before cochlear implantation. Overall, there was no statistically significant improvement in SRT70 when comparing the original HAs with new replacement HAs. Most participants did not improve their SRT70 while keeping the task engagement constant throughout the HINT, as demonstrated with pupillometry. For most participants, replacing the HAs prior to CI surgery did not affect speech perception. However, some participants experienced relevant improvements with the new replacement HAs, and, therefore, might have postponed the CI surgery as a result. Decisions regarding CI candidacy should be made on a case-by-case basis. Furthermore, if the replacement HAs are compatible with a CI and fitted in a bimodal solution, they may provide additional benefits. However, further research is needed to explore the long-term benefits of new replacement HAs fitted with a CI in a bimodal solution.

## Figures and Tables

**Figure 1 audiolres-15-00013-f001:**
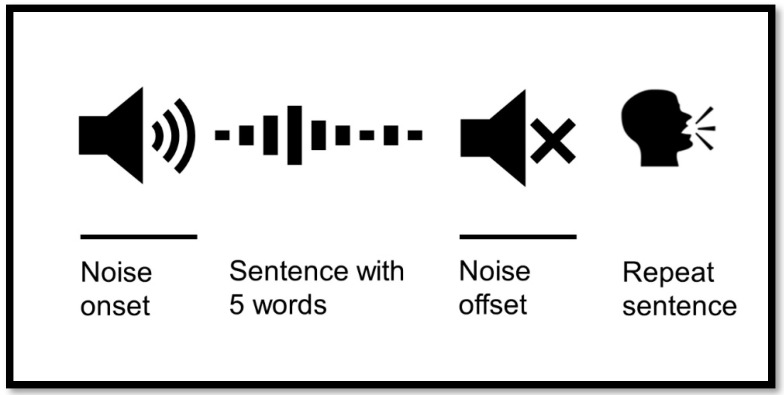
Hearing in Noise Test (HINT) setup with pupillometry. The noise onset appears before the signal and continues after signal offset. The signal is a sentence with 5 words in Danish. The participant repeats the sentence after the noise ends.

**Figure 2 audiolres-15-00013-f002:**
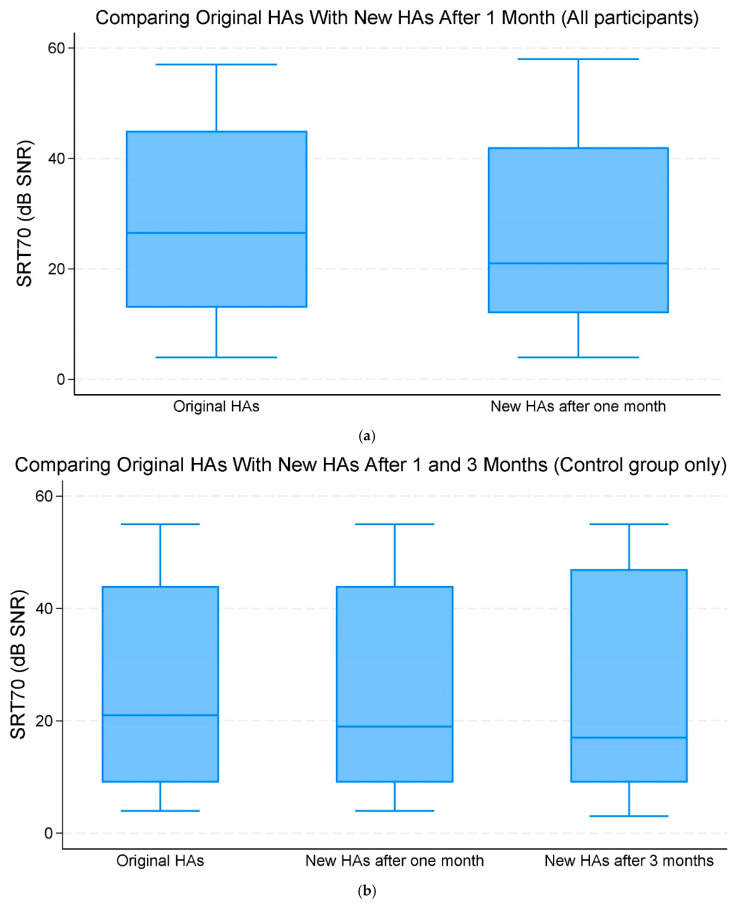
(**a**) The median and interquartile range (IQR) of speech reception threshold (SRT)70 (70% correct word recognition) (dB signal-to-noise ratio (SNR)). Comparing the original hearing aids (HAs) with the new HAs after one month of use (all participants). (**b**) The median and IQR of SRT70 dB SNR. Comparing the original HAs with the new HAs after one and three months of use (control group only).

**Figure 3 audiolres-15-00013-f003:**
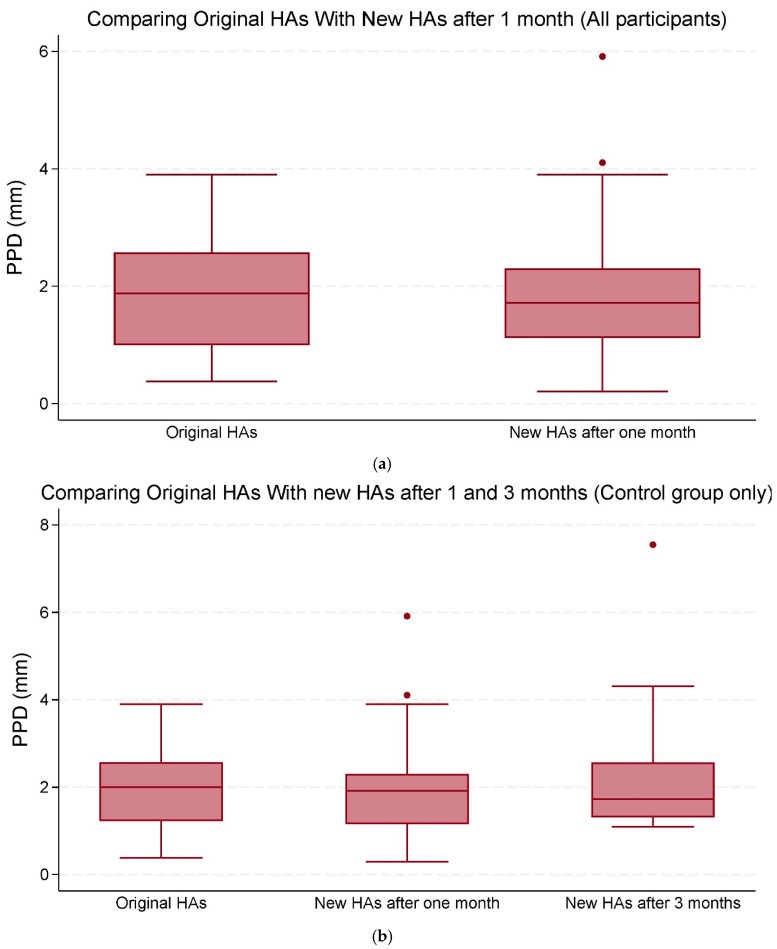
(**a**) The median and interquartile range (IQR) of peak pupil dilation (PPD) (mm). Comparisons were made between the original hearing aids (HAs) and the new HAs after one and three months of use (all participants). (**b**) The median and IQR of PPD (mm). Comparisons were made between the original HAs and the new HAs after one and three months of use (control group only). The data points outside the boxes represent outliers.

**Figure 4 audiolres-15-00013-f004:**
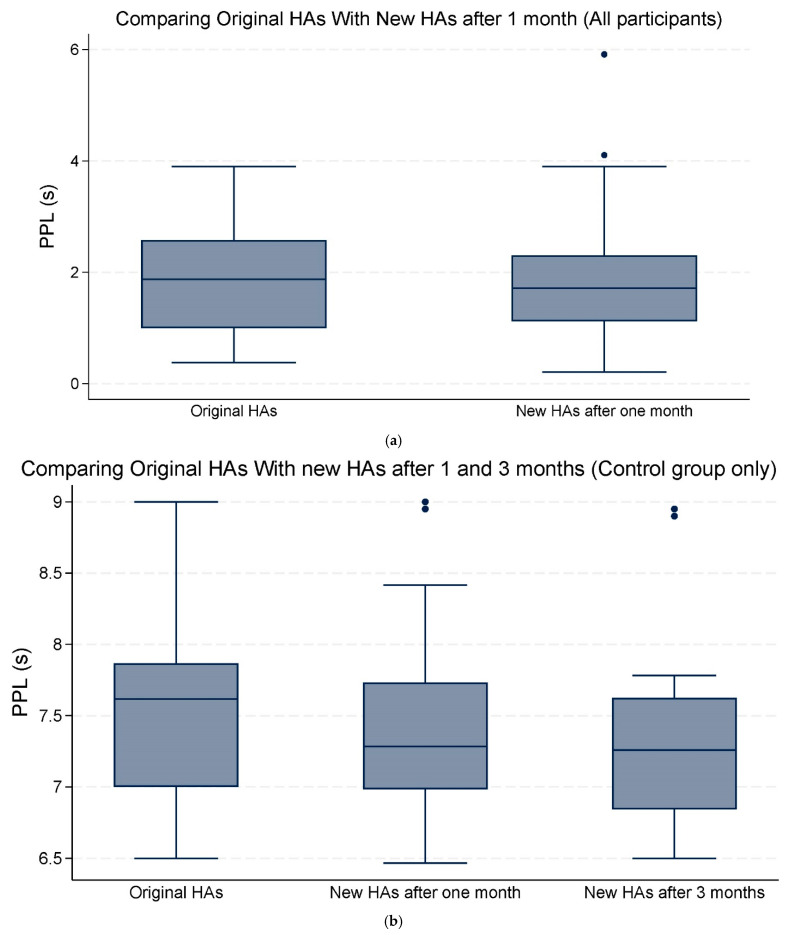
(**a**) The median and interquartile range (IQR) of peak pupil latency (PPL) seconds (s). Comparing the original hearing aids (HAs) with the new HAs after one month of use (all participants). (**b**) The median and IQR of PPL (s). Comparing the original HAs with the new HAs after one and three months of use (control group only). The data points outside the boxes represent outliers.

**Table 1 audiolres-15-00013-t001:** Participant Characteristics Across Intervention and Control Group by Follow-Up Time.

		Original HAs (n = 50)		New HAs After One Month (n = 50)		New HAs After Three Months (n = 25)	
Age (yrs) mean ± SD		63.4 ± 17.2		63.4 ± 17.2		67.4 ± 13.0	
Range		23–82		23–82		38–81	
Female (%)		44.0		44.0		32.0	
(PTA-min) ^1^ (dB HL), mean ± SD		68.6 ± 21.8		68.6 ± 21.8		64.9 ± 25.6	
(PTA-max) ^2^ (dB-HL), mean ± SD		90.6 ± 17.5		90.6 ± 17.5		91.6 ± 18.0	
		PTA-min	PTA-max	PTA-min	PTA-max	PTA-min	PTA-max
Normal Hearing (≤19 dB HL)		5.3	NA	3.1	NA	7.1	NA
Mild (20–34 dB HL)		8.0	NA	9.2	NA	17.9	NA
Moderate (35–49 dB HL)		4.0	NA	3.1	NA	NA	NA
Moderate-Severe (50–64 dB HL)		24.0	4.9	26.2	5.4	28.6	3.9
Severe (65–79 dB HL)		20.0	21.3	23.1	23.2	17.9	23.1
Profound (≥80 dB HL)		38.7	73.8	35.4	71.4	28.6	73.1
SRT70	Mean ± SD	28.5 ± 16.0		26.6 ± 16.0		24.7 ± 17.5	
(dB SNR)	Median (IQR)	26.5 (13.0, 45.0)		21.0 (12.0, 42.0)		17.0 (9.0, 47.0)	
PPD (mm)	Mean ± SD	1.8 ± 0.9		2.0 ± 1.4		2.4 ± 1.9	
	Median (IQR)	1.9 (1.0, 2.6)		1.7 (1.1, 2.3)		1.7 (1.3, 2.6)	
PPL (s)	Mean ± SD	7.6 ± 0.8		7.5 ± 0.7		7.4 ± 0.8	
	Median (IQR)	7.7 (7.0, 8.0)		7.4 (7.0, 7.8)		7.3 (6.8, 7.6)	

^1^ PTA-min, Pure-Tone Average for the better ear, HL, Hearing Loss. ^2^ PTA-max, Pure-Tone Average for the ear considered for CI. Pure-Tone Average of 0.5, 1, 2, and 4 kHz. N number, SD standard deviation, IQR Interquartile range, SNR Signal-to-Noise-Ratio, dB Decibel, PPD Peak Pupil Dilation and PPL Peak Pupil Latency.

**Table 2 audiolres-15-00013-t002:** Mixed regression analysis of SRT70 dB SNR and adjusted SRT70 ≤ 20 dB SNR. Comparing the original HAs with the new HAs after one month of use (all participants), and comparing the original HAs with the new HAs after one and three months of use within the control group only.

	All Participants				Control Group Only			
	SRT70 ^1^ (dB SNR ^2^) (N = 50)		SRT70 ≤ 20 (dB SNR) (N = 50)		SRT70 (dB SNR) (N = 27) (After 1 Month) (N = 25) (After 3 Month)		SRT70 ≤ 20 dB SNR (N = 27) (After 1 Month) (N = 25) (After 3 Month)	
	*Coef. (95%CI)*	*p-Value*	*Coef. (95%CI)*	*p-Value*	*Coef. (95%CI)*	*p-Value*	*Coef. (95%CI)*	*p-Value*
**Original HAs vs. new HAs after 1 month**	−1.9 (−4.69;0.89)	0.182	−0.38 (−1.24;0.48)	0.387	−0.67 (−4.31;2.98)	0.720	−0.74 (−1.99;0.51)	0.244
**Original HAs vs. new HAs after 3 months**	NA	NA	NA	NA	−0.56 (−4.41;3.28)	0.773	−0.46 (−1.78;0.85)	0.490
**Sex** **(Female)**	4.25 (−2.64;11.14)	0.227	1.84 (−0.33;4.01)	0.096	3.89 (−4.25;12.03)	0.349	0.10 (−2.80;3.01)	0.946
**Age**	0.13 (−0.08;0.33)	0.223	0.08 (0.01;0.14)	**0.016**	0.18 (−0.08;0.44)	0.178	0.08 (−0.02;0.17)	0.101
**PTA-min ^3^**	0.36 (0.24;0.49)	**<0.001**	0.15 (0.11;0.19)	**<0.001**	0.39 (0.27;0.52)	**<0.001**	0.15 (0.04;0.05)	**<0.001**
**PTA-max ^4^**	0.03 (−0.08;0.15)	0.596	−0.001 (−0.04;0.03)	0.947	0.05 (−0.08;0.18)	0.457	0.01 (−0.04;0.05)	0.716

^1^ Speech reception threshold 70. ^2^ Sound-in-noise ratio. ^3^ PTA-min, Pure-Tone Average for the better ear, HL, Hearing Loss. ^4^ PTA-max, Pure-Tone Average for the ear considered for CI. Pure-Tone Average of 0.5, 1, 2 and 4 kHz. Significant *p*-values marked in bold.

**Table 3 audiolres-15-00013-t003:** Mixed regression analysis of pupillometric outcomes as PPD (mm) and PPL (s). Comparing the original HAs with the new HAs after one month of use (all participants), and comparing the original HAs with the new HAs after one and three months of use within the control group only.

	All Participants				Control Group			
	PPD ^1^ (mm) (N = 35)		PPL ^2^ (s) (N = 35)		PPD (mm) (N = 20)		PPL (s) (N = 20)	
	*Coef. (95%CI)*	*p-Value*	*Coef. (95%CI)*	*p-Value*	*Coef. (95%CI)*	*p-Value*	*Coef. (95%CI)*	*p-Value*
**Original HAs vs. new HAs after 1 month**	0.17 (−0.30;0.64)	0.476	−0.11 (−0.46;0.23)	0.510	0.14 (−0.51;0.78)	0.677	0.003 (−0.44;0.45)	0.991
**Original HAs vs. new HAs after 3 months**	NA	NA	NA	NA	0.35 (−0.51;0.78)	0.346	−0.13 (−0.64;0.37)	0.610
Sex (Female)	0.07 (−0.62;0.76)	0.844	−0.06 (−0.46;0.33)	0.756	0.30 (−0.42;1.01)	0.420	0.05 (−0.39;0.48)	0.827
Age	−0.01 (−0.03;0.01)	0.237	0.002 (−0.01;0.01)	0.779	−0.03 (−0.06;−0.002)	**0.038**	−0.004 (−0.02;0.01)	0.625
PTA-min ^3^ (dB HL)	0.01 (−0.01;0.02)	0.420	0.004 (−0.01;0.01)	0.317	0.01 (−0.001;0.02)	0.071	0.003 (−0.005;0.01)	0.475
PTA-max ^4^ (dB HL)	0.03 (0.01;0.05)	**0.002**	0.001 (−0.01;0.01)	0.844	0.02 (0.004;0.04)	**0.016**	0.01 (−0.002;0.02)	0.116

^1^ Peak pupil dilation. ^2^ Peak pupil latency. ^3^ Pure-Tone Average for the better ear. ^4^ Pure-Tone Average for the ear considered for CI. Pure-Tone Average of 0.5, 1, 2 and 4 kHz. Significant *p*-values marked in bold.

## Data Availability

The data is shared as metadata/codebook from Redcap Open Patient data Explorative Network (OPEN). All data files is stored in OPEN analyze. All identifiers were removed before analysis. Data will be shared through publications as well. The data management plan was uploaded to the website: https://dmponline.deic.dk in 1 February 2022.

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
