# Peer review of "Evaluation of Replacement Hearing Aids in Cochlear Implant Candidates Using the Hearing in Noise Test (HINT) and Pupillometry"

_audiolres, 2025, doi:10.3390/audiolres15010013_

Round 1
Reviewer 1 Report
Comments and Suggestions for Authors
Summary: Thanks for submitting this work for review. I had difficulty seeing the rationale for the work, as it is presented in the introduction. There are competing rationales, one of which relates to evaluating candidacy status and one which relates to benefit of bimodal stimulation post implant. It is never clear which of these is motivating the current work until reaching the conclusion (seems to be evaluating candidacy?). The methods need extension work as they are not sufficient for replication, in my opinion. Due to formatting, Table 2 is not readable. Below are specific comments. I think that by really expanding the methods to be very specific and detailed would go a long way in improving the paper and allowing for possible publication.
Page 1 Line 38: What makes switching to a new HA difficult? I think this could be expanded upon, also, why is it necessary to switch to a new HA? Or why is it recommended?
Lines 41 – 48: I don’t see how this section relates to the study until reaching line 105. It needs to be made clear to the reader that pre-CI eval necessitates use of a different hearing aid than what the listener may be accustomed to. There should be some discussion on reasons this may be disadvantageous to the patient (if it would be…I’m not convinced that this is an issue…if anything, I might expect the patient to do poorer and be more likely to meet CI candidacy). I think the rationale is weak, as it is currently written. There is so much focus on pre-candidacy eval. Is the concern for meeting candidacy or is the concern for after implantation? If the latter, there needs to be much stronger rationale as to why the patient can’t continue using their old hearing aids.
Lines 78: Stated that PPD and PPR increase with task difficulty and that “more profound hearing loss causes the listening effort to increase thus resulting in an increase in PPD.” The citations provided support the relationship between PPD and task difficulty, but not the bit about “more profound hearing loss” causing increase effort. The primary purpose of studies 10 and 19 was on task difficulty and PPD. Task difficulty was manipulated by degrading the speech. These studies didn’t examine degree of loss.
Explanation is needed as to how the SRT70 was measured.
Lines 173 – 179: This is very unclear. What is the “first HINT” and what is “a fixed SRT70”?
How can you be sure the listener was able to detect the noise onset? Also, its stated that the noise masker had to be less than 45 dB to ensure it was detectable. Should this have been “greater than 45 dB”?
For “SNR” in Table 1 – is this the SNR for SRT70?
The formatting of Table 2 makes it very difficult to understand the data. This needs to be reformatted to allow readability.
There needs to be better explanation on the methods/rationale for setting the SRT70 to 20 dB SNR (line 254). There is a tendency in this paper for methodological information to show up in the results. It makes it difficult to follow what the design of the study was as well as the rationale behind the design.
Lines 298 – 300 – I don’t follow…pupillometry was analyzed in the results but it is now stated that it was impossible. Please explain.
Comments on the Quality of English LanguageNA
Author Response
|
Response to Reviewer 1 Comments
|
||
|
1. Summary |
|
|
|
Thank you very much for taking the time to review this manuscript. Please find our detailed responses below, along with the corresponding revisions highlighted in track changes within the re-submitted files.
|
||
|
3. Point-by-point response to Comments and Suggestions for Authors |
||
|
Reviewer 1 Summary: Thanks for submitting this work for review. I had difficulty seeing the rationale for the work, as it is presented in the introduction. There are competing rationales, one of which relates to evaluating candidacy status and one which relates to benefit of bimodal stimulation post implant. It is never clear which of these is motivating the current work until reaching the conclusion (seems to be evaluating candidacy?). The methods need extension work as they are not sufficient for replication, in my opinion. Due to formatting, Table 2 is not readable. Below are specific comments. I think that by really expanding the methods to be very specific and detailed would go a long way in improving the paper and allowing for possible publication. Response: Thank you for your thorough review of our manuscript. We agree that the rationale was not as clear as it could be, and we have made significant revisions to the introduction to address this. The main purpose of the study is to evaluate the change of hearing aids as a part of the preoperative assessment of cochlear implant candidates. Hopefully, this stands out more clear in the current revision. Additionally, we expanded the methods section to provide more detail, ensuring greater transparency and replicability. The tables have also been revised to improve clarity and assist the reader. Comments 1: Page 1 Line 38: What makes switching to a new HA difficult? I think this could be expanded upon, also, why is it necessary to switch to a new HA? Or why is it recommended? Response 1: Thank you for pointing this out. We agree with this comment. Therefore, we have revised the manuscript introduction, the change can be found on – page 3-4, line 60-83. Here we give some arguments for the advantage of switching hearing aids.
Comments 2: Lines 41 – 48: I don’t see how this section relates to the study until reaching line 105. It needs to be made clear to the reader that pre-CI eval necessitates use of a different hearing aid than what the listener may be accustomed to. There should be some discussion on reasons this may be disadvantageous to the patient (if it would be…I’m not convinced that this is an issue…if anything, I might expect the patient to do poorer and be more likely to meet CI candidacy). I think the rationale is weak, as it is currently written. There is so much focus on pre-candidacy eval. Is the concern for meeting candidacy or is the concern for after implantation? If the latter, there needs to be much stronger rationale as to why the patient can’t continue using their old hearing aids. Response 2: Thank you for this comment, we agree that our rationale could be described better and therefore we have made a thorough revision to ensure greater clarity. To address this we have revised a large part of the introduction from line 60-96. See also response 1. Comments 3: Lines 78: Stated that PPD and PPL increase with task difficulty and that “more profound hearing loss causes the listening effort to increase thus resulting in an increase in PPD.” The citations provided support the relationship between PPD and task difficulty, but not the bit about “more profound hearing loss” causing increase effort. The primary purpose of studies 10 and 19 was on task difficulty and PPD. Task difficulty was manipulated by degrading the speech. These studies didn’t examine degree of loss. Response 3: Thank you for bringing this to our attention. The sentence was not intended to reference "profound hearing loss," and we appreciate your comment on the misquotation. We have removed the part related to profound hearing loss. The changes are in page 6, line 141-144. Comments 4: Explanation is needed as to how the SRT70 was measured. Response 4: Thank you very much for this comment. We recognized that the paragraph on the measurement of SRT70 was inadequate and have revised the entire "Apparatus and Procedure" section (pages 10-12, lines 233-267, pages ) to give a better description on how SRT70 was measured. Comments 5: Lines 173 – 179: This is very unclear. What is the “first HINT” and what is “a fixed SRT70”? Response 5: Thank you very much for this comment. As mentioned in response 4, we recognize that the paragraph regarding the two sessions of HINT and the fixed/adaptive SRT70 required further clarification. The paragraph has been extensively revised to give an explanation of the HINT setups and the fixed SRT70.
Comments 6: How can you be sure the listener was able to detect the noise onset? Also, its stated that the noise masker had to be less than 45 dB to ensure it was detectable. Should this have been “greater than 45 dB”? Response 6: Thank you for pointing this out. It is absolutely correct, and we have changed this accordingly to be greater than 45 dB – see page 12 line 266. Comments 7: For “SNR” in Table 1 – is this the SNR for SRT70? Response 7: Thank you for this comment. We have added SRT70 in parentheses next to SNR to clarify that these refer to the same outcome. Comments 8: The formatting of Table 2 makes it very difficult to understand the data. This needs to be reformatted to allow readability. Response 8: Thank you for your comment. We have made revisions to the table to improve its readability. Comments 9: There needs to be better explanation on the methods/rationale for setting the SRT70 to 20 dB SNR (line 254). There is a tendency in this paper for methodological information to show up in the results. It makes it difficult to follow what the design of the study was as well as the rationale behind the design. Response 9: We completely agree with this comment and have made significant revisions to the method paragraph. It is described in the paragraph “Apparatus and Procedure” in lines 233 to 267 the reason for setting the SNR70 to 20 dB. The noise needs to be audible with hearing aids on, and this is the reason for not reducing the intensity of the noise signal further.
Comments 10: Lines 298 – 300 – I don’t follow…pupillometry was analyzed in the results but it is now stated that it was impossible. Please explain. Response 10: Thank you for highlighting this incorrect statement. Pupillometry was not impossible to conduct; however, some patients were unable to perform the test due to their profound hearing loss, which prevented them from hearing the noise masker—a necessary component for conducting the pupillometry test. The changes are in page 20, line 471-475.
|
||
|
|
||
|
|
||
|
|
||
|
|
||

Reviewer 2 Report
Comments and Suggestions for Authors
Subtitles in the abstract such as methods (Line 19), and conclusions, are not seen.
Patient characteristics and etiologies should be given. And the mean age was over 60 with a wide range (23-83), cognitive assessments should be done. If it is done, the results should be reported.
Line 165: why just half of the participants were tested 3 months after? It is intentional or not? This should be specified.
Table 1 is very complicated and includes very much data. Dividing the table into two parts could be clearer. PTA should be given separately for each ear.
Maybe further analysis can be done with different age groups.
For comparisons done with original HA, 1-month and 3-month usage. The number of participants in the third month decreased. Selecting the same participants and comparing them (28 subjects) according to usage could produce different results.
Different hearing loss degrees are also limitations. This should be stated and discussed in the manuscript.
Author Response
|
Response to Reviewer 2 Comments
|
||
|
1. Summary |
|
|
|
Thank you very much for taking the time to review this manuscript. Please find our detailed responses below, along with the corresponding revisions highlighted in track changes within the re-submitted files. |
||
|
3. Point-by-point response to Comments and Suggestions for Authors Comments 1: Subtitles in the abstract such as methods (Line 19), and conclusions, are not seen. Response 1: Thank you for your comment. Several subtitles have been added. Comments 2: Patient characteristics and etiologies should be given. And the mean age was over 60 with a wide range (23-83), cognitive assessments should be done. If it is done, the results should be reported. Response 2: Thank you for your comment. As a result, we have added the PTA for the poorer hearing ear in Table 1. Unfortunately, we did not conduct any cognitive assessments but patients have to be able to cooperate to all the performed examinations.
Comments 3: Line 165: why just half of the participants were tested 3 months after? It is intentional or not? This should be specified. Response 3: Thank you for your comment. We recognized that we had not adequately described the participants' grouping after one month of hearing aid acclimatization. Consequently, we have provided a detailed explanation of the two groups, clarifying why half of the participants were tested three months later. Page 13, line 293-303.
Comments 4: Table 1 is very complicated and includes very much data. Dividing the table into two parts could be clearer. PTA should be given separately for each ear. Maybe further analysis can be done with different age groups. Response 4: Agreed. We have revised both tables and included the PTA for the poorer hearing ear as well. We do not think it is appropriate to do analyses with different age-groups as this will give very small groups if we have to divide the groups further. However, age is included in the data analyses, so age is controlled for (see paragraph Data analyses for further details).
Comments 5: For comparisons done with original HA, 1-month and 3-month usage. The number of participants in the third month decreased. Selecting the same participants and comparing them (28 subjects) according to usage could produce different results. Response 5: Thank you for this comment. Mixed regression can accommodate situations where there are two groups with unequal numbers of subjects. Mixed regression is well-suited for analyzing data with groups of unequal sizes. Its flexibility makes it a robust choice for handling complex data structures, even when the group sizes differ significantly. We are therefore confident that this problem has been handled.
Comments 6: Different hearing loss degrees are also limitations. This should be stated and discussed in the manuscript. Response 6: Thank you very much for this comment. We have revised the paragraph on limitations to address this important topic. The varying degree of hearing loss is indeed a crucial factor in the improvement of hearing performance with CI and HA in a bimodal solution. See additions at page 20-21, line 465-479.
|
||

Reviewer 3 Report
Comments and Suggestions for Authors
This is an interesting and well-written manuscript investigating whether speech perception in noise remains unchanged or improves with replacement HAs compared to original HAs in CI candidates with residual hearing. The analyses are well presented and the manuscript is easy to read.
Here are my comments:
lines 44-45: Since there is no statistically significant difference between the original and the new CI-compatible HAs, improvement of meant SRT70 is mot different from no improvement. Please consider replacing improvement with change.
Line 71 - change Cis to CIs
Lines 319 - 327 -- Consider replacing improvement with change for all statistically not significant changes in thresholds. looking at the word "improvement" followed by statistically not significant results makes it difficult to understand.
Lines 344 - 346 -- No statistics was provided for significant association between PPD and PTA-max
Line 355: please expand CORS as it is introduced for the first time.
Lines 355-358 -- the authors have identifed the use of biCROS HAs as the reason for SNR70 improvement. were there any commanalities among the others who showed improvement but werent using biCROS HAs? If so, that needs to be addressed here.
The authors note that "..can be more difficult to get a sufficient pupil response in older adults". The authors could try using proportionate change in pupil diameter as the dependent measure instead of peak pupil dilation. This might show differences in the pupil change as a function of task difficulty.
Also, please report effect sizes for all statistical comparisons. This would be really helpful as the authors note that some individuals experience relevant improvements with the new hearing aids and there was
Comments on the Quality of English LanguageThe quality of English is good.
Author Response
Response to Reviewer:
Thank you very much for taking the time to review this manuscript. Please find our detailed responses below, along with the corresponding revisions highlighted in track changes within the re-submitted files. Please see the attachment with the response-letter in word file.
Reviewer:
This is an interesting and well-written manuscript investigating whether speech perception in noise remains unchanged or improves with replacement HAs compared to original HAs in CI candidates with residual hearing. The analyses are well presented and the manuscript is easy to read.
Here are my comments:
Comments 1: Lines 44-45: Since there is no statistically significant difference between the original and the new CI-compatible HAs, improvement of meant SRT70 is mot different from no improvement. Please consider replacing improvement with change.
Response 1: Thank you for this comment, which we agree will make the results paragraph more clear, so we replaced “improvement” with “change”.
Comments 2: Line 71 - change Cis to Cis
Response 2: Thank you for this comment.
Comments 3: Lines 319 - 327 -- Consider replacing improvement with change for all statistically not significant changes in thresholds. looking at the word "improvement" followed by statistically not significant results makes it difficult to understand.
Response 3: Thank you! We want the paper to be easy to follow, so we replaced “improvement” with “change” to enhance clarity.
Comments 4: Lines 344 - 346 -- No statistics was provided for significant association between PPD and PTA-max
Response 4: Yes, thanks very much, we have now provided statistics.
Comments 5: Line 355: please expand CORS as it is introduced for the first time.
Response 5: Thank you, the text has been revised so the abbreviation biCROS has been placed in a parenthesis as it is presented the first time.
Comments 6: Lines 355-358 -- the authors have identifed the use of biCROS HAs as the reason for SNR70 improvement. were there any commanalities among the others who showed improvement but werent using biCROS HAs? If so, that needs to be addressed here.
Response 6: We know that some patients are referred to cochlear implantation with poorly fitted HAs. Therefore we revised the text as follows:
Some patients referred for a CI were not optimally fitted with the original HAs, potentially achieving better speech perception had their original HAs been appropriately fitted. Among these, some utilized bilateral microphones with contralateral routing of signal (biCROS) HAs, which after using new bilateral HAs were provided amplification to the previously unaided ear. Following the fitting of new HAs, these individuals reported subjective improvements in sound localization and enhanced speech understanding. This observation suggest that new replacement HAs may improve speech intelligibility for certain patients, even though the changes were not statistically significant at the group level.
Comments 7: The authors note that "..can be more difficult to get a sufficient pupil response in older adults". The authors could try using proportionate change in pupil diameter as the dependent measure instead of peak pupil dilation. This might show differences in the pupil change as a function of task difficulty.
Response 7: Thank you very much for this comment.The proportionate change will be included, as we have used a mixed regression model for statistical analyses where the random effects are attributed to the individuals (See Table 3). This means, that the change in pupil response between different measurements is related to the individual differences (random effects) as well as the group differences (fixed effects). An alternative strategy could be to use the proportionate change as the outcome and then omit the time variable, which has been included as fixed effect in the mixed regression analyses. Mixed regression models are often useful for longitudinal data, and we have decided to keep this model.
Comments 8: Also, please report effect sizes for all statistical comparisons. This would be really helpful as the authors note that some individuals experience relevant improvements with the new hearing aids and there was
Response 8: Thank you for this very important comment, we have provided the effect sizes for all statistical comparisons.

Round 2
Reviewer 1 Report
Comments and Suggestions for Authors
I found the introduction much improved. The background and rationale for the project is conveyed clearly.
The methods section has also been improved as has presentation of the results. In the first version, it was very difficult to understand what was being measured, design of the study, and the results. The current version makes these clearer. Unfortunately, I think there are some significant problems with the study and approach:
1) Lines 213-222: What is the "measured value of the SRT70"? Is this the level of the noise babble or the SNR?
2) Lines 213-222: Just making sure I understand regarding the SRT70 > 20 dB...So, if the SNR at SRT70 was 25 dB, the fixed test had to occur at 20 dB b/c speech was fixed at 65 and noise needed to be at least 45 dB for audiblity...correct?
3) Lines 277 - 284:
What is the "mean SRT70"?
Do you mean "mixed effects"? If so, please clarify what the fixed and random effects were.
4) Results: This is one of my biggest concerns. It seems that you have taken the data and divided into 3 "groups": Baseline (original HA), 1-month w/new HA, and 3-months w/new HA. You then do all of your stats and comparisons using the same baseline group, despite that group including subjects who weren't in the 1-month group. This seems incorrect. Shouldn't there be 2 baseline groups? - A baseline group for the 1-month new HA subjects (which may be all of them as it sounds like everyone was tested at 1 month) and then a second baseline group for the 3-month new HA subjects? Additionally, when comparing the 1-month HA group to the 3-month HA group, the 1-month group should only include the same subjects who participated in the 3-month group (n=28). I don't have confidence in how the groups were formed and whether they allow for meaningful comparisons.
Another major concern I have is how you handled the data from individuals with SRT70 > 20 dB. Why not throw this data out? Perform analysis that includes only valid data (SRT70 for SNRs <= 20 dB). This should be the only analysis that is performed as it is the only one that makes sense. Including data where SRT70 > 20 dB is a problem b/c of your design...not being able to reach a "steady state during HINT"...what does this mean? Does this mean the SRT70 is not accurate? If so, that data should be not be included in the analysis. Acknowledge the problem and then remove it from analysis. Similarly, thresholding the SRT70 > 20 dB to 20 dB is also a problem. This is meaningless data. It looks like half of your data typically has SRT70 > 20 dB (medians reported in table 1). This is a problem.
Author Response
Audiology Research
Manuscript ID: audiolres-3209441
Title: Evaluation of Replacement Hearing Aids in Cochlear Implant Candidates
Using the Hearing in Noise Test (HINT) and Pupillometry
Authors: Yeliz Jakobsen *, Kathleen Faulkner, Lindsey Van Yper, Jesper Hvass Schmidt
*Corresponding author
Dear Editor-in-Chief,
We are grateful for the opportunity to resubmit our article to Audiology Research and would like to extend our sincere thanks to you and the reviewers for your valuable and constructive feedback. We believe that your suggestions have significantly improved the quality of our manuscript.
In response to the reviewers’ comments, we have addressed each one in detail. For clarity, we have copy-pasted the original questions and comments, numbered them consecutively, and provided our corresponding responses “Response:” below. All revisions to the manuscript have been made using the “Track Changes” function.
Yeliz Jakobsen, Doctor, PhD Fellow
Department of Otorhinolaryngology, Head and Neck Surgery and Audiology,
Institute of Clinical Research
University of Southern Denmark, Odense, Denmark
Phone: +4530669135
Email: Yeliz.Jakobsen@rsyd.dk
|
Response to Reviewer 1 Comments
|
||
|
1. Summary |
|
|
|
Thank you very much for taking the time to review this manuscript again. Please find our detailed responses below, along with the corresponding revisions highlighted in track changes within the re-submitted files.
|
||
|
3. Point-by-point response to Comments and Suggestions for Authors |
||
I found the introduction much improved. The background and rationale for the project is conveyed clearly.
The methods section has also been improved as has presentation of the results. In the first version, it was very difficult to understand what was being measured, design of the study, and the results. The current version makes these clearer. Unfortunately, I think there are some significant problems with the study and approach:
- Lines 213-222: What is the "measured value of the SRT70"? Is this the level of the noise babble or the SNR?
Response: Thank you for this comment, we changed the paragraph as follows: “The speech signal was set at 65 dB, and the noise masker level was adjusted to each participant's individual SNR, as determined in the first HINT session. If the adaptive procedure in HINT found an SRT70 below 20 dB SNR, this measured SRT70 value was used as the fixed level for the subsequent HINT session. However, if the adaptive procedure found an SRT70 above 20 dB SNR, the SRT70 was limited to +20 dB SNR in the following session to ensure the noise onset and offset remained audible.” (P.10, l. 227-232)
So the measured level of the SRT70 will always be a signal to noise ratio, and as the level of noise will be set according to the measured SNR. In case of SNR of 20 dB, the speech signal will be at 65 dB and the noise level at 45 dB.
2) Lines 213-222: Just making sure I understand regarding the SRT70 > 20 dB...So, if the SNR at SRT70 was 25 dB, the fixed test had to occur at 20 dB b/c speech was fixed at 65 and noise needed to be at least 45 dB for audiblity...correct?
Response: Yes, that is correct. If the participant's SRT70 score was 23 dB SNR in the first HINT session, the SRT70 was adjusted to 20 dB SNR. In the subsequent HINT session, the speaker level was set to 65 dB, with the noise speakers at 45 dB.
3) Lines 277 - 284:
What is the "mean SRT70"?
Do you mean "mixed effects"? If so, please clarify what the fixed and random effects were.
Response: Thank you very much for this question and we believe that with this paragraph it is clarified:
“The study hypothesized that speech perception would remain unchanged with new replacement HAs compared to the original HAs in CI candidates with residual hearing. A constrained linear mixed model was used to test this hypothesis, including the groups (original HAs, new HAs after one month, and new HAs after six months), time points (baseline and follow-up), and their interaction as fixed effects. Patient ID was included as a random effect to account for repeated measurements. Covariates such as age, sex, and pure-tone average (PTA) thresholds for both the poorer-hearing ear (PTA-max) and the better-hearing ear (PTA-min) were also included.”
(p. 13, l. 291-297)
4) Results: This is one of my biggest concerns. It seems that you have taken the data and divided into 3 "groups": Baseline (original HA), 1-month w/new HA, and 3-months w/new HA. You then do all of your stats and comparisons using the same baseline group, despite that group including subjects who weren't in the 1-month group. This seems incorrect. Shouldn't there be 2 baseline groups? - A baseline group for the 1-month new HA subjects (which may be all of them as it sounds like everyone was tested at 1 month) and then a second baseline group for the 3-month new HA subjects? Additionally, when comparing the 1-month HA group to the 3-month HA group, the 1-month group should only include the same subjects who participated in the 3-month group (n=28). I don't have confidence in how the groups were formed and whether they allow for meaningful comparisons.
Another major concern I have is how you handled the data from individuals with SRT70 > 20 dB. Why not throw this data out? Perform analysis that includes only valid data (SRT70 for SNRs <= 20 dB). This should be the only analysis that is performed as it is the only one that makes sense. Including data where SRT70 > 20 dB is a problem b/c of your design...not being able to reach a "steady state during HINT"...what does this mean? Does this mean the SRT70 is not accurate? If so, that data should be not be included in the analysis. Acknowledge the problem and then remove it from analysis. Similarly, thresholding the SRT70 > 20 dB to 20 dB is also a problem. This is meaningless data. It looks like half of your data typically has SRT70 > 20 dB (medians reported in table 1). This is a problem.
Response: Thank you very much for this comment. The results paragraph has been thoroughly revised and we agree with you that it may not have been meaningful to do the comparisons between groups as we dig in the previous version of the manuscript. The study population was divided into two groups: one analysis was conducted across all participants, and another was done within the control group. The only difference of the control group was, that this group had measurements at an extra time point at three months. In this version of the analyses we have excluded the subjects with missing data. Five participants had missing data at 1 month follow-up with new HAs which resulted in the necessity to run the analysis again with updated tables and figures. The figures are separated into “a” and “b” to ensure accurate representation of sample sizes for different follow-up periods. Thus, A Figures relate to whole study population with measurements at baseline and 1 months follow-up, whereas B Figures relate to the control group with measurements at baseline, 1 month and 3 months follow-up. The control group is included in the study population with measurements at baseline and at one month follow-up. Additionally, the tables have been updated with the updated sample sizes.
An additional analysis was conducted with the study participants measured with SRT70 >20dB to ensure that there was no statistically significant difference when leaving out subjects with SRT70 values below 20dB SNR. The SRT70 data are still meaningful. The only reason for the concern related to an SNR above 20dB is related to the pupillometry tests. We can demonstrate that it has no significant impact of the pupillometry results t include subjects where SRT70 has been measured with an SNR above 20dB.
